# Design and Analysis of the Dual-Band Far-Field Super-Resolution Metalens with Large Aperture

**DOI:** 10.3390/nano14060513

**Published:** 2024-03-13

**Authors:** Cheng Guo, Zhishuai Zheng, Ziang Liu, Zilong Yan, Yucheng Wang, Ruotong Chen, Zhuonan Liu, Peiquan Yu, Weihao Wan, Qing Zhao, Xiaoping Huang

**Affiliations:** 1School of Resources and Environment, University of Electronic Science and Technology of China, Chengdu 611731, China; guocheng@uestc.edu.cn (C.G.); 202311070408@std.uestc.edu.cn (Z.Z.); 202221070109@std.uestc.edu.cn (Z.L.); lza202221070109@163.com (Z.Y.); 202222070313@std.uestc.edu.cn (Y.W.); 202122070206@std.uestc.edu.cn (R.C.); 202322070205@std.uestc.edu.cn (Z.L.); 15663667723@163.com (P.Y.); wanweihao0306@163.com (W.W.); zhaoq@uestc.edu.cn (Q.Z.); 2Yangtze Delta Region Institute (Huzhou), University of Electronic Science and Technology of China, Huzhou 313001, China

**Keywords:** dual band, super-resolution metalens, large aperture, far-field imaging

## Abstract

The resolving power of metalens telescopes rely on their aperture size. Flat telescopes are advancing with the research on super-resolution confocal metalenses with large aperture. However, the aperture sizes of metalenses are usually bound within hundreds of micrometers due to computational and fabrication challenges, limiting their usage on practical optical devices like telescopes. In this work, we demonstrated a two-step designing method for the design of dual-band far-field super-resolution metalens with aperture sizes from the micro-scale to macro-scale. By utilizing two types of inserted unit cells, the phase profile of a dual-wavelength metalens with a small aperture of 100 μm was constructed. Through numerical simulation, the measured FWHM values of the focal spots of 5.81 μm and 6.81 μm at working wavelengths of 632.8 nm and 1265.6 nm were found to all be slightly smaller than the values of 0.61 λ/NA, demonstrating the super-resolution imaging of the designed metalens. By measuring the optical power ratio of the focal plane and the incident plane, the focusing efficiencies were 76% at 632.8 nm and 64% at 1265.6 nm. Based on the design method for small-aperture metalens, far-field imaging properties through the macro metalens with an aperture of 40 mm were simulated by using the Huygens–Fresnel principle. The simulation results demonstrate confocal far-field imaging behavior at the target wavelengths of 632.8 nm and 1265.6 nm, with a focal length of 200 mm. The design method for dual-band far-field super-resolution metalens with a large aperture opens a door towards the practical applications in the dual-band space telescope system.

## 1. Introduction

As a crucial component of optical confocal systems, dual-wavelength confocal lenses have been extensively explored. However, traditional lens sets designed for wavelength confocalization are often bulky and expensive, posing obstacles to the integrated, planar, and multifunctional development of modern optical systems [1]. Telescopes play an important role in astronomy, serving as vital tools for exploring the night sky. However, the resolving power of telescopes largely depends on their aperture size, which results in significant weight and footprint, posing challenges to their construction and deployment [2]. Light Detection and Ranging (LiDAR) technology is an active remote sensing system that is utilized in autonomous vehicles, machine vision, and augmented reality. LiDAR technology operates by measuring the time it takes for a short-pulsed light to scatter back from a target object and return to a detector. This measured time is then converted to distance based on the speed of light. By scanning illumination across a target scene, the system reconstructs a three-dimensional (3D) image [3]. Traditional optical systems rely on complex multi-element designs to achieve the desired levels of optical performance, but components made of traditional refractive materials like glass are both bulky and heavy. Moreover, these components are often cascaded together for aberration and distortion correction, leading to a sharp increase in system size and weight. One potential solution to these issues may lie in the so-called metalens technology, which can mold light using subwavelength nanoantennas that vary in a planar space. The ultra-thin planar metalenses have garnered significant attention, promising substantial reductions in the size and weight of optical systems [2].

Over the last decade, metalenses have been proposed, possessing the revolutionary property to overcome the diffraction limit by capturing the high-level spatial frequency components of an object [1,4,5,6,7,8,9,10,11,12,13]. Metalens composed of subwavelength metamaterials or metasurfaces can image and resolve the subwavelength details, thus achieving super-resolution imaging [5,6,7,8,9]. By utilizing the integration of traditional optics and modern metasurface technology, metalens exhibits remarkable features, including free optical adjustment, tremendous design flexibility, precise control, smaller volume, and lighter weight [1,5,6,7,8,9,10,11,12,13]. These attributes of metalens provide significant promising applications in the realms of optical fiber imaging, microscopic imaging, and photographic lenses [13,14]. The basic working principle of a metalens involves the use of subwavelength scatterers to tailor the characteristics of light that carry subwavelength information from the objects. By precisely controlling the scattered light, the metalens extract fine details and convert them into propagating waves, achieving super-solution focusing and imaging [4,5,6]. Dielectric metalenses, in particular, comprise a quasi-periodic array of subwavelength-sized dielectric nanoantennas with spatially varying geometries to modulate scattered light [15,16,17,18]. They can offer particular cutting-edge lensing functions owing to low loss, arbitrary wavefront encoding, strong dispersion, polarization tunability, high efficiency, and novel integral compatibility [17,18]. Despite the promising capabilities of metalenses, challenges remain [19,20], including a narrow operation band, high loss, and limited scalability. Mostly, the working frequency of a metalens relies on the resonance of its unit, making it work in a narrow band [21]. The realization of a large-scale achromatic band with high performance still remains a big challenge [22,23].

Confocal metalens allows for multi-wavelength laser light to transmit signal, which increases signal transmission capacity and reliability. At present, a variety of pioneering works have demonstrated multi-wavelength confocal metalens research schemes over a broad wavelength range, from visible to near-infrared [24,25,26,27,28,29,30,31,32]. Generally, these achromatic metalenses are successfully realized by dispersive phase compensation [25], spatial multiplexing [26], or multispectral achromatic phase transition [27]. However, there are still many challenges towards the practical achromatic confocal applications, such as relatively complex structure and relatively low focusing efficiency, especially for the metalens with macro-scale aperture [22]. Metasurface research has rapidly evolved into a new phase, incorporating a variety of new concepts and exploring new materials for a variety of interdisciplinary applications. One of the most interesting is the advent of the metalens. In 2016, Capasso’s research group at Harvard University reported that the metalens has the advantages of large size, high efficiency, and large numerical aperture [7]. Its single-wavelength imaging near the diffraction limit in the visible band is no less effective than that of traditional optical lenses, and its advantages of thin and compact proportions are expected to replace traditional imaging lenses in people’s lives. In 2017, Din Ping Tsai’s group proposed a design principle to realize achromatic metasurface devices, which successfully eliminated chromatic aberration over a continuous wavelength region from 1200 to 1680 nm for circularly polarized incidences in a reflection scheme [33]. Based on their approach, broadband achromatic metasurface devices have significantly innovated a whole class of designs in the field of full-color light manipulation and imaging, pushing metasurface devices towards practical applications in full-color light manipulation and imaging. In 2022, Tao Li proposed and experimentally realized non-ideal achromatic flat lenses with a large-scale diameter of up to 10 mm, employing a two-step method from the micro-scale to the macro-scale [34]. Therefore, further developments are required in the design and fabrication of broadband, large-aperture long-focal-length achromatic metalens to meet the demands of remote astronomical observation and telecommunications in advanced optical systems [2,35,36,37].

In this work, we employ a novel two-step design method to design dual-band far-field super-resolution metalens with aperture sizes from the micro-scale to macro-scale by utilizing two types of inserted unit cells. The hybrid designing principle was strategically utilized by tuning the Pancharatnam–Berry (P–B) geometrical phases and additional phases for each meta-atom. To achieve the super-resolution metalens working in dual-band, the two neighboring circularly positioned nanorods were inserted with different radius and heights. Herein, the discrete P–B phase was controlled by the integrated resonant elements in the metalens. Based on the design method for small-aperture super-resolution metalens, the macro-scale metalens with a large aperture and long-focal-length was further designed by constructing a macro-sized nanorod array. According to the waveguide theory, the effective medium theory, and the generalized Snell’s law, the propagation and focusing characteristics of the proposed planar metalens with a diameter of 40 mm were simulated and calculated, which were found to work in two central wavelengths of 632.8 nm and 1265.6 nm with a bandwidth of 100 nm. By using the Huygens–Fresnel principle, far-field imaging through the metalens with a focal length to aperture ratio of 5 was also simulated to study the focusing properties. Our work proposes a design of metalens that simultaneously works in dual-band confocal style while reducing wavelength crosstalk and improving focusing efficiency. The comprehensively proposed metalens design method with improved optical performance represents a significant step towards the practical applications of remote detecting and imaging in the dual-band space telescope system.

## 2. Device Design

### 2.1. The Principle of Device Design

To achieve a dual-band confocal working mode, the two-frequency wavefront phase of the metalens should be designed carefully within the multiple nanocolumn arrays [4]. In order to converge the incident light ray at two different wavelengths to the same focal point, the phase regulation provided by the metalens needs to meet the hyperbolic phase controlling formula [37]:(1)$$\varphi_{m}(x,y)=\frac{2\pi }{\lambda_{m}}(f-\sqrt{x^{2}+y^{2}+f^{2}}),$$
where *m* is the number of incident waves, $\lambda_{m}$ is the target wavelength, f is the target focal length, and (*x*, *y*) is the coordinate position of the metalens microelement structure. Therefore, precisely adjusting the phase offset by means of the generalized Snell’s law and transmission phase principle is the key to focusing the dual-frequency band light to the same point by the designed metalens. The multiple-nanorod array provides the phase control in the metalens, and the introduced phase offset is expressed as follows [32]:(2)$$\varphi_{m}=\frac{2\pi }{\lambda_{m}}n_{eff}H,$$
where *n_eff_* is the effective refractive index of the unit structure and H represents the propagation distance. When the wavelength and the height of the nanostructures are fixed, the equivalent refractive index *n_eff_* can be changed by varying the lattice constants of the grating P and the radius of the nanostructure R to achieve phase accumulation covering [0, 2π].

The superposition radiation field in the focal plane of *z* = *f* can be expressed by the following formula [38]:(3)$$E(r,\lambda )=\frac{2\pi }{i\lambda }\frac{e^{ik(f+\frac{x^{2}}{2f})}}{z}\int_{0}^{a}(J_{0}(\frac{kxr}{z}))rdr,$$
where *f* is the focal length, *k* is a constant of wave number representing the beamwidth factor, and λ is the incident light wavelength. *J_0_* is the Bessel function of zero order, *r* = (x^2^ + y^2^)^1/2^ is the radial coordinate in the Fourier plane, and a is the radial coordinate in the aperture plane. It can be observed that for a metalens at any wavelength, the field intensity at any point on its focal plane is determined by the radius and focal length of the metalens. Based on the light ray-controlling principle, the design of a large-aperture metalens is proposed to achieve long focal length and macroscopic far-field radiation. Herein, the wavefront of an incident wave was engineered with a suitably designed amplitude-phase mask to achieve an arbitrarily small point spread function (PSF). This opens up another avenue for far-field optical super-resolution. Based on the design principle, a novel super-resolution metalens composed of large-area multiple-nanocolumn arrays was designed, which can achieve dual-band far-field con-focusing with high focusing efficiency.

### 2.2. Confocal Imaging Model of the Dual-Band Metalens with Long Focal Length

To demonstrate a large-area dual-band metalens with the same long focal length of 200 mm, we designed a confocal metalens that was 40 mm in diameter. TiO_2_ nanorod arrays on SiO_2_ were chosen as the material for the metalens (Figure 1a). In order to produce this confocal lensing effect, the metalens imposed a spatial phase profile on the wavefront of $\varphi =2\pi /\lambda (f-\sqrt{r^{2}+f^{2}})$, where *λ* = 632.8 nm, *λ* = 1253.6 nm is the design wavelengths, *r* is the radial coordinate, and *f* = 200 mm is the focal length (Figure 1b). The phase profile focusing collimated light into a spot was constructed using the dense nanorod array, each of which acted as a miniature antenna to locally impart a controlled phase shift. Due to their polarization-independent response and smaller surface area, the nanorods were chosen as the meta-elements to design the metalens, providing better control over light polarization and improving transmission efficiency. To achieve macroscopic far-field transmission and confocal imaging at two wavelengths, we strategically controlled the radius and height of the nanorod array to meet the phase requirements of the entire metalens. Generally, by varying the diameter, the nanorods with fixed height were able to produce 2π phase coverage and high, relatively uniform transmittances. To achieve confocal imaging with two wavelengths, the heights of the nanorods were varied alternately and extended from the center to outward, as shown in Figure 1b.

Based on this principle, the dual-band confocal metalens was designed through searching for the best phase coverage and high, relatively uniform transmittances with the optimized diameter and height of the nanorod. Firstly, the phase coverage and transmittance of the nanorod were scanned with the S parameter method at 632.8 nm, wherein the diameter of the nanorod ranged from 100 nm to 360 nm, and the height ranged from 1.45 μm to 1.7 μm. Each parameter was scanned with 11 data points, resulting in 121 data points in total. The phase diagram and transmittance diagram at different diameters and heights were calculated and analyzed, as shown in Figure 2a,b, respectively. It can be seen that when the height is about 1.6 μm, the phase covers from −π to π, and the transmittance is mostly above 0.8, which meets the phase coverage and transmittance requirement for a metalens. Similarly, the phase and transmittance diagram were also calculated at 1265.6 nm with the diameter ranging from 40 nm to 440 nm, and the height ranging from 1.5 μm to 2.5 μm. The obtained phase and transmission diagram are shown in Figure 2c,d. It can be seen that when height is about 2.2 μm, the phase covers from −π to π, and the transmittance is mostly close to 1, which meets the phase coverage requirement and transmittance requirement. In the following metalens design, a lattice constant of 500 nm between adjacent nanorods was adopted to meet the optical efficiency requirement and avoid neighboring cross-talk. Thus, the dual-band confocal metalens was designed with a nanorod array working at two wavelengths with a bandwidth of 100 nm, wherein some nanorods (height H = 1.6 μm, diameters ranging from 100 nm to 360 nm) and other nanorods (height H = 2.2 μm, diameters ranging from 40 nm to 440 nm) were placed alternately from the center of the circle to outwards in a circular shape. Because of the memory space limitation, the numerical simulation was carried out first using the finite difference time domain (FDTD) method for smaller metalens (D = 100 μm) with a short focal length. Then, based on the same phase profile construction principle, a large-area metalens (D = 40 mm) with long focal length of 200 mm was designed and simulated using the Huygens–Fresnel method.

## 3. Result and Discussion

### 3.1. Dual-Wavelength Metalens with Small Aperture

To prove of our design concept, a small-aperture metalens was demonstrated with a diameter of D = 100 μm at working wavelengths of 632.8 nm and 1265.6 nm. To verify the broadband behavior of the metalens, linear polarized light with a bandwidth of 100 nm in each wavelength was presented on the designed device. Here, we numerically show the focusing properties of the designed metalens by using the FDTD method. Figure 3a,b depicts the electric field intensity distribution in the x–z plane and x–y plane at the focal point. As shown in Figure 3c, the focal length was about 0.85 mm for the working wavelength of 632.8 nm. The focusing resolution of the designed metalens was measured by the full width at the half maximum (FWHM) of the focusing spot. As shown in Figure 3d, the calculated FWHM was 5.81 μm at the working wavelength of 632.8 nm. For the working wavelength of 1265.6 nm, the imaging properties of the same metalens were also simulated. As shown in Figure 4a,b, the electric field distribution in the x–z plane and x–y plane at the focal point are illustrated. According to our calculation, the focal length in Figure 4c is about 0.55 mm for the working wavelength of 1265.6 nm. As shown in Figure 4d, the calculated FWHM is 6.81 μm at the working wavelength of 1265.6 nm. The small difference in the focal length at the two working wavelengths may be attributed to the significant deviation in the phase shift brought by the identical phase profile of a small-aperture metalens. Fortunately, this deviation drawback in focal length was corrected in the following designed large-aperture metalens with a diameter of 40 mm.

To compare, we can obtain the theoretical resolution limit by calculating the numerical aperture (NA). According to the diameter of 100 μm and the obtained focal length of the designed metalens, the numerical apertures were calculated as NA_632.8nm_ = 0.0555 and NA_1265.6nm_ = 0.0996 for two working wavelengths. Putting the NA values into the resolution limit formula R = 0.61 λ/NA of the metalens, the theoretical resolution limits of the metalens are R_632.8_ = 6.95 μm and R_1265.6_ = 7.74 μm, respectively, for the two working wavelengths. Obviously, the calculated FWHM values of the spot size through the designed metalens are slightly smaller than the values of 0.61 λ/NA for the two working wavelengths. The trade-off with the very small NA values contributes to the enhancement in the resolution. By measuring the optical power ratio of the focal plane and the incident plane, the focusing efficiencies are 76% at 632.8 nm and 64% at 1265.6 nm. Thus, the super-resolution of the designed metalens is realized at the two working wavelengths [34].

### 3.2. Dual-Wavelength Metalens with Large Aperture and Long Focal Length

Because of the memory space limitation of the FDTD simulation method, it is very difficult to directly simulate a metalens with a large macroscopic aperture (>10 mm order) and long focal length. To realize the simulation over macroscopically large device areas, we have implemented a two-step algorithm that generates these design files. Firstly, the relationship between the phase profile of the small-aperture (100 μm) metalens and the radius of the nanorod element were obtained by the FDTD method. Then, based on the construction principle of the phase profile of the small-aperture metalens, the target phase profile matrix required by the large-aperture (40 mm) imaging system was generated, and the far-field image was calculated using the Huygens–Fresnel principle.

For the design of a microstructure imaging device with an aperture of 40 mm and focal length of 200 mm, we first used the formula $\varphi \left(r\right)=\varphi \left(0\right)-\frac{2\pi }{\lambda_{d}}\left(\sqrt{\left(r^{2}+f^{2}\right)}-f\right)$ to generate a phase profile. The aperture of the phase profile is 40 mm, and the target phase period of each cell structure is 500 nm, which were used to generate the phase profile with wavelengths of 632.8 nm and 1265.6 nm, respectively, and each profile contained 80,000 data points, as shown in Figure 5. Figure 5a,b shows the phase profile and the magnified profile at the center area of the designed metalens at 632.8 nm, respectively. Similarly, Figure 5c,d shows the phase profile and the magnified profile at the center area of the designed metalens at 1265.6 nm, respectively.

Based on the generated phase profile, the Huygens–Fresnel principle was used to write an algorithm to calculate the far-field electric field distribution. Here, an observation imaging matrix of 5 cm × 25 cm was calculated for far-field imaging, and each matrix was divided into 99 × 99 observation matrices. Each observation matrix was used to perform far-field imaging based on the phase profile, ultimately contributing to the formation of the far-field electric field distribution of the large-area metalens. As shown in Figure 6a,b, the far-field electric field distribution of a single-frequency metalens with two separate working wavelengths of 632.8 nm and 1265.6 nm was generated, with the focal length reaching 200 mm.

To achieve confocal imaging, the influence of the non-working wavelengths on the metalens should be introduced. By inserting 4000 random number matrices into the phase profile of the single-frequency metalens, a new phase matrix containing 80,000 data points was generated. Based on this method, an algorithm was written to calculate the far-field electric field distribution by using the Huygens–Fresnel principle. Thus, the phase profile of the nanorod array was obtained by using the radius and phase curve data. Using the above method, the far-field electric field distributions of the dual-band metalens with working wavelengths of 632.8 nm and 1265.6 nm were generated, with a focal length of 200 mm, as shown in Figure 6c,d. Due to the existence of random sequences, the intensity of the focal spot appears darker than that in a single-frequency metalens. Exactly, the confocal imaging was demonstrated in the dual-band metalens with a large aperture of 40 mm. It is worth mentioning that this work is aimed at the application of LiDAR. The two working wavelengths were decided with a double relation according to the special requirement.

## 4. Conclusions

In summary, we demonstrate a novel two-step method to design dual-band far-field super-resolution metalens with aperture sizes ranging from the micro-scale to macro-scale. Based on the design method for small-aperture super-resolution metalens, 200 mm far-field imaging properties through the macro metalens with an aperture of 40 mm were simulated by using the Huygens–Fresnel principle. By utilizing two types of inserted unit cells, the dual-wavelength metalens with a small aperture of 100 μm was designed and simulated. The measured FWHM values of the focal spot of 5.81μm and 6.81 μm at working wavelengths of 632.8 nm and 1265.6 nm are all smaller than the theoretical value of 0.61 λ/NA, demonstrating the super-resolution imaging of the designed metalens. The high focusing efficiencies of 76% at 632.8 nm and 64% at 1265.6 nm demonstrate the optimization of the metalens structure. Based on the design method for small-aperture metalens, a dual-wavelength metalens with a large aperture of 40 mm was simulated by further using the Huygens–Fresnel principle. The numerical simulation results demonstrate its confocal far-field imaging behavior at the target wavelengths, with a focal length of 200 mm. The results of our work provide a new way for designing dual-band far-field super-resolution metalens with large aperture towards the practical applications of the space telescope system.

## Figures and Tables

**Figure 1 nanomaterials-14-00513-f001:**
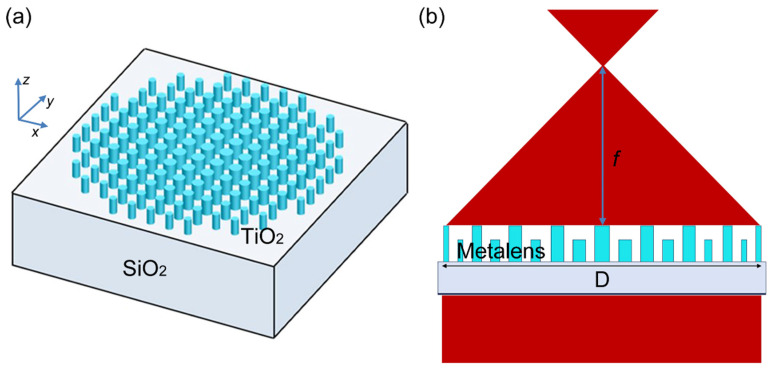
Metalens design. (**a**) A schematic shows a metasurface lens (metalens) designed to focus light of normal incidence, where D is the diameter and f is the focal length. (**b**) The schematic of the dual-band confocal metalens composed of the nanorod array with the heights and diameters varying alternately and extending from the center to outwards.

**Figure 2 nanomaterials-14-00513-f002:**
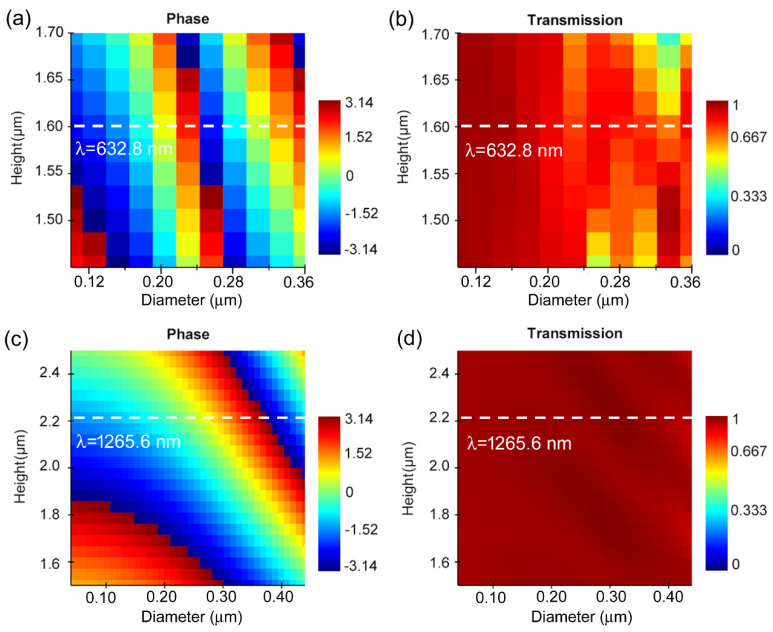
The obtained phase and transmittance diagrams of the nanorod with the change in diameter and height according to the analysis of S parameter. (**a**) The phase diagram at a wavelength of 632.8 nm. (**b**) The phase diagram at a wavelength of 1265.6 nm. (**c**) The transmittance diagram at a wavelength of 632.8 nm. (**d**) The transmittance diagram at a wavelength of 1265.6 nm.

**Figure 3 nanomaterials-14-00513-f003:**
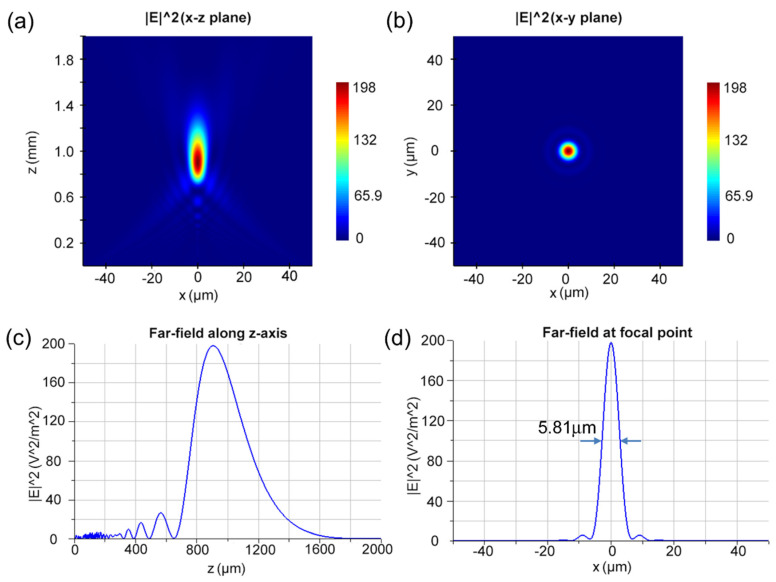
(**a**,**b**) The electric field distribution diagram of x–z plane and x–y plane at the working wavelength of 632.8 nm. (**c**) Far-field along z-axis at a wavelength of 632.8 nm. (**d**) The electric field distribution at focal plane at a wavelength of 632.8 nm, wherein the FWHM is 5.81 μm.

**Figure 4 nanomaterials-14-00513-f004:**
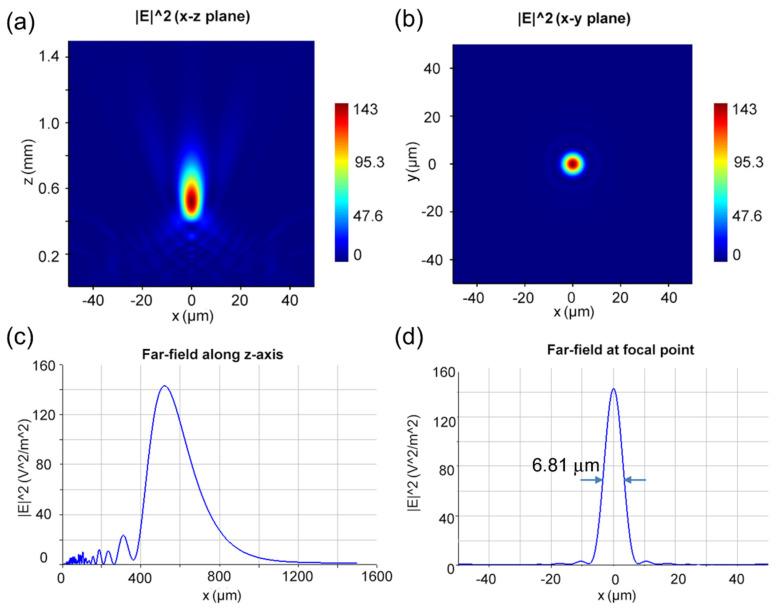
(**a**,**b**) The electric field distribution diagram of the x–z plane and x–y plane at the working wavelength of 1265.6 nm. (**c**) Far-field along z-axis at a wavelength of 1265.6 nm. (**d**) The electric field distribution at the focal plane at a wavelength of 1265.6 nm, wherein the FWHM is 6.81 μm.

**Figure 5 nanomaterials-14-00513-f005:**
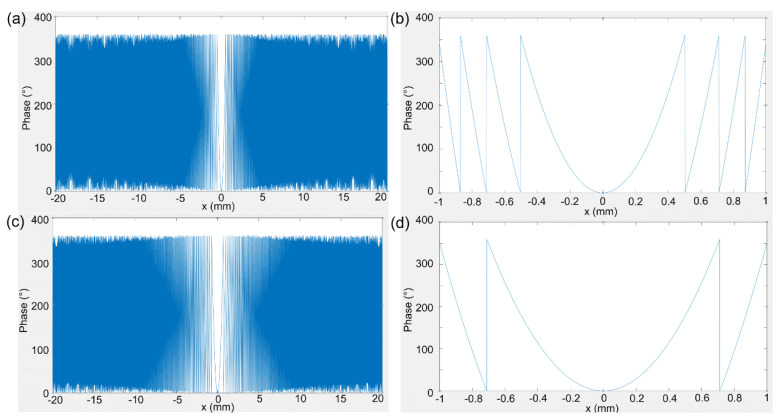
The phase profile as a function of the radius of the designed metalens with a large aperture of 40 mm. (**a**,**b**) The phase profile and the magnified profile at the center area of the designed metalens at 632.8 nm. (**c**,**d**) The phase profile and the magnified profile at the center area of the designed metalens at 1265.6 nm.

**Figure 6 nanomaterials-14-00513-f006:**
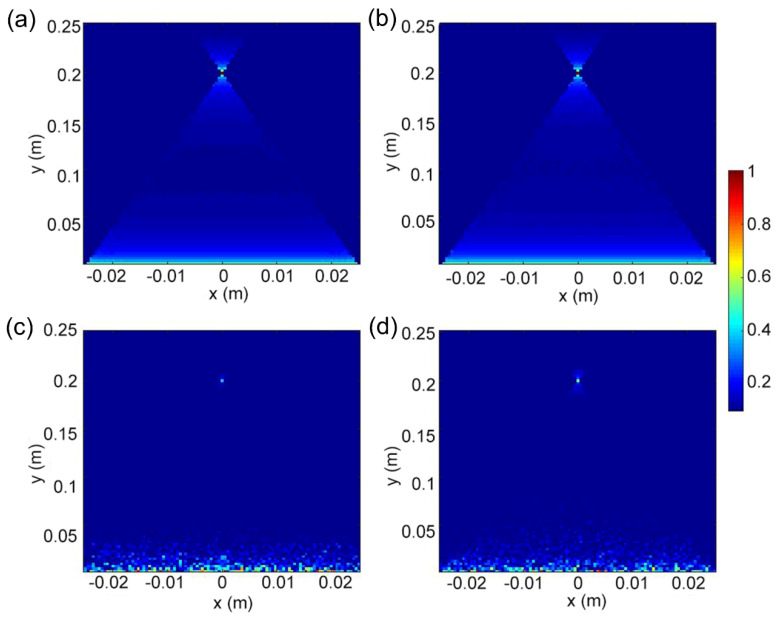
(**a**,**b**) Electric field distribution diagram through the single-frequency metalens at working wavelength of 632.8 nm and 1265.6 nm, respectively. (**c**,**d**) Electric field distribution diagram of dual-band metalens working at wavelengths of 632.8 nm and 1265.6 nm.

## Data Availability

The data presented in this study are available on request from the corresponding author.

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
