# Peer review of "Design and Analysis of the Dual-Band Far-Field Super-Resolution Metalens with Large Aperture"

_nanomaterials, 2024, doi:10.3390/nano14060513_

Round 1

Reviewer 1 Report

Comments and Suggestions for Authors

The manuscript describes a metalens design that can achieve focusing at two wavelengths simultaneously. While the topic is interesting and the steps taken look logical, the overall description lacks many important details. The reviewer thinks that the manuscript can be reviewed only after significant revisions.

1. The design method, which combined Huygens-Fresnel principle and FDTD simulations, look very standard. If there were any novel contribution from the authors, they need to emphasize it proactively.

2. The choice of the two "working" wavelengths seems too artificial. The authors tried 632.8 nm and 1265.6 nm, an exact integer multiple or the original. Was the choice related to the design method proposed? Can the method work in other wavelength pairs? If so, the authors are advised to show the results in the revision.

3. All the technical details of the simulation processes are missing. The authors simulated materials with over 600 nm wavelength interval. The reviewer got concerned if the dispersion effect were properly considered. All other computation-specific details were sparse at best.

4. Some of the plots were given without any real information. Figure 5 shows VERY rapidly varying phase values that are lumped into a thick line, without giving any information. Figure 6 plots were rendered at very low resolution with the important details just hidden by a vast non-informative background. These need to be improved.

5. Overall, the abstract, introduction, and conclusion commonly lack the authors' "novelty claim" paragraphs or sentences. The novelty and significance of the work must be emphasized properly.

6. No comments were given on the realization of the design.   

Comments on the Quality of English Language

Some minor proofreading and editing required.

Reviewer 2 Report

Comments and Suggestions for Authors

This manuscript is devoted to development of two-step method to design dual-band far-field super-resolution metalens with aperture size from micro-scale to macro-scale. The simulation of characteristics of dual-wavelength metalens with two types of inserted unit cells (e.g., full width at half maximum values of the focal spot) and far-field imaging properties through the macro metalens by using Huygens-Fresnel principle. The topic of manuscript is important for scientific groups in areas of electrodynamics, for technological groups developing the metastructures etc .

There are some points to make the information more clear or to correct some details:

1)      It is necessary to explain the abbreviation at first mention. Some abbreviations (e.g. P-B phase for Pancharatnam-Berry phase probably (68th line) are without explanation.

2)      It is necessary to explain all values in formulas. There are some symbols without this explanation (e.g., neff and H in (2), r, a, J0 in (3)). Reader should not guess what  the radius-vector or polynomial is. It is not clear to obtain these formulas by authors or to take from other sources. There are not the references before or after (2) and (3).

3)      The symbol before m in 100 “unknown symbol“m (213th line) is unknown. It should be the “m” for milli- or “m” for micro-, probably.

4)      Authors presented the results of simulations of field in various planes obtained at design simulated previously in this manuscript. Could authors give the comparison with any real results obtained in literature, possibly?

This manuscript is written sufficiently clear and describes with details the design of dual-band far-field super-resolution metalens and some characteristics for such systems obtained by digitally simulation.

The manuscript can be published after minor revisions.

Round 2

Reviewer 1 Report

Comments and Suggestions for Authors

The reviewer thinks the manuscript is ready for publication.

Comments on the Quality of English Language

The reviewer thinks the manuscript is ready for publication.